# Strategies to Facilitate Interorganizational Collaboration in County-Level Opioid Overdose Prevention and Response: A Qualitative Analysis

**DOI:** 10.3390/ijerph22121765

**Published:** 2025-11-21

**Authors:** Julia Dickson-Gomez, Sarah Krechel, Jessica Ohlrich, Jennifer Hernandez-Meier, Constance Kostelac

**Affiliations:** 1Institute for Health and Humanity, Medical College of Wisconsin, Milwaukee, WI 53226, USA; skrechel@mcw.edu (S.K.); johlrich@mcw.edu (J.O.); ckostelac@mcw.edu (C.K.); 2Department of Emergency Medicine, Medical College of Wisconsin, Milwaukee, WI 53226, USA; jhernandez@mcw.edu

**Keywords:** opioid use disorder, overdose, harm reduction, interorganizational collaboration, community-based

## Abstract

Community-level overdose prevention interventions often require collaboration among organizations from various sectors including emergency medicine, criminal justice, harm reduction, and drug treatment organizations, yet little is known about ways to foster interorganizational collaboration among organizations with very different missions and in different socio-political contexts. This paper presents results from interviews with key informants involved in overdose prevention coalitions in two counties in Wisconsin (n = 45). Key informants were purposively selected from 31 different organizations in sectors including harm reduction, drug treatment, emergency medicine, and law enforcement. Interviews asked participants to describe the overdose crisis in their communities and the work they do, including any partnerships or coalitions formed with other organizations. We conducted thematic analysis using inductive and deductive coding. Participants’ experiences illuminate strategies and actions that facilitated coalitions’ work (interorganizational processes) and changed the context in which they worked to be more accepting of harm reduction efforts and less stigmatizing and punitive toward people who use opioids (PWUO). These included getting the word out in community-facing events to educate the public and destigmatize harm reduction, working with representatives across the CoC in various sectors, and actively working with them to create shared missions. Key people acted as bridges while others had the power to convene multiple agencies to a common cause. Overdose Fatality Reviews (OFRs) were found to be particularly helpful in identifying gaps in the current Opioid CoC and developing programs in collaboration with other organizations to address them. Organizational empowerment offers a useful framework for understanding how to facilitate IOC at the intra- (e.g., community education to reduce stigma, inter- (bridging roles by key actors), and extra-organizational levels (e.g., policy changes supporting naloxone access). These strategies can be used by coalition members and tested in future community-level overdose responses.

## 1. Introduction

Opioid overdose deaths in the United States remain a public health crisis with over 100,000 deaths per year since 2021, peaking at 107,941 overdose deaths in 2022 with a slight decrease in 2023 [1,2]. Wisconsin largely mirrors national trends both in terms of overdoses peaking in 2022 at 1459 and decreasing to 1415 in 2024 [3]. This rise in overdoses is largely driven by synthetic opioids such as fentanyl and its analogs although increasingly stimulants are involved with opioids both nationally and in Wisconsin [4]. The causes and consequences of opioid overdose are multiple and complex, and overdose rates are especially high among those with multiple intersecting stigmas including substance use, race, homelessness, mental health, and legal needs [5]. Individual organizations are often not equipped to meet all clients’ needs, leading many communities to establish coalitions to address the many facets of the opioid crisis [6]. These coalitions are built around organizations that make up the Opioid Continuum of Care (CoC) or are situated at various points of the Sequential Intercept Model (SIM).

A large body of research has shown that medications to treat opioid use disorder (MOUD), including buprenorphine, methadone, and naltrexone, are the “gold standard” for treating opioid use disorder [7]. Further, research suggests that people on buprenorphine for six or more months are much less likely to suffer a fatal overdose [8,9,10]. The CoC, therefore, consists of seven stages to link people with opioid use disorder (OUD) into long-term MOUD treatment. These include: 1. Estimation of affected population; 2. Prevention of OUD among those at risk; 3. Diagnosis among those affected with OUD. 4. Linkage to OUD care among those diagnosed. 5. Medication (MOUD) initiation among those entering care. 6. Retention (e.g., for at least six months) among those initiating MOUD; 7. Remission or recovery among those retained on MOUD [11]. The CoC requires coordination from various agencies and health systems for timely information about opioid overdoses in the community to provide sufficient information for planning purposes. It may also include organizations that supply harm reduction or psychosocial services to provide the support needed for treatment success [7].

The SIM was developed from the observation that people with behavioral health issues, including those with substance use disorders, come into frequent contact with the criminal justice system and that the various intercept points could be opportunities to link people to drug treatment or harm reduction [12]. Programs have been developed at multiple intercepts to divert people to treatment such as pre-arrest programs, post-arrest programs, drug treatment courts, and connection to treatment or harm reduction while incarcerated or under community supervision (probation and parole) [12]. People with OUD who have experienced non-fatal overdoses similarly frequently encounter emergency medical personnel such as through emergency medical services (EMS) and the emergency department (ED), and programs have been designed to link these individuals to drug treatment (warm handoffs) or to initiate buprenorphine in the ED [7,13,14]. All of these require collaboration among EMS, and community MOUD treatment and law enforcement is involved in these efforts in some jurisdictions. Many programs also include social workers to link to the needed social services, and peer support specialists to navigate the system and provide support from the perspective of someone with lived experience [13,14].

More recently, Overdose Fatality Reviews (OFRs) have emerged as an important component of community-level overdose prevention efforts. OFRs are modeled after child fatality reviews (CFRs) or hospital-based fatality reviews. OFRs include members of diverse disciplines and organizations that may have interacted with the decedent during their lifetime or have subject matter expertise on the functioning of various systems. The teams review overdose deaths to investigate risk factors and circumstances leading to the death and recommend interventions to prevent such deaths in the future. OFRs were initially established in several jurisdictions in Wisconsin starting in 2016. Since 2016, additional teams have been established primarily with pass-through federal funding granted by the Wisconsin Department of Health Services and the Wisconsin Department of Justice. The training and technical assistance for local teams has been provided by the Medical College of Wisconsin, Inc. [15].

The fact that these programs require high levels of interorganizational collaboration (IOC) was not lost on major funders including the Centers for Disease Control and Prevention (CDC), U.S. Department of Justice (DOJ), and National Institutes of Health (NIH), which have required communities to form coalitions and to use overdose data in prioritizing overdose interventions. For example, the NIH Helping to End Addiction Long-term (HEAL) Initiative requires community coalitions and the development of data dashboards to inform interventions. Further, those awarded are required to adopt evidence-based practices that address the overdose, including overdose education and naloxone distribution (OEND), effective delivery of MOUD, and safer opioid prescribing and dispensing, all of which are enhanced by effective IOC [16].

Despite the perceived importance of IOC, limited effort has been made to evaluate the success, quality, or effectiveness of IOCs that have been funded through these efforts. For example, HEAL qualitative research of the community–hospital collaborations suggested that while community members valued collaboration with hospitals, hospitals sometimes disengaged with coalitions due to conflicting goals, interests, cultures, and even languages [6]. The Juvenile Justice Translational Research on Interventions for Adolescents in the Legal Systems aimed to improve service delivery and integrated care for youth in the juvenile justice system by creating collaborations between juvenile justice, social services, and drug treatment. The research compared two implementation strategies to improve service delivery, but did not include measures of IOC [17]. Creating a strategy to increase something we have not yet agreed how to measure leads to the inevitable question of “How would we know if the strategy worked or not?”

Part of the difficulty in evaluating IOC is the diversity of perspectives by which IOC can be understood. IOC can be seen as a pattern or exchange of information, resources, people, or clients to achieve improved health outcomes, and social network approaches are often used to measure and evaluate IOC [18,19]. For example, a social network analysis was used to characterize IOC among networks that had formed Accountable Communities of Health (ACH). The ACH model convenes community stakeholders to develop systematic screening and referral systems to address the gap in providing coordinated clinical care, public health, and social services. Results of the study suggested that the ACH was effective in stimulating connection across sectors. Factors associated with sustainable connections included strength of relationships and type of collaboration, namely data and resource sharing [20].

Research has thus focused on factors that encourage these exchanges. Factors found in the literature that lead to successful collaborations include aligned interests or shared mission [21,22,23,24], the presence of people who act as bridges [21,25], psychological ownership [22], provisional teamwork and learning [21], and trust and strength of relationships [22,23,24]. People serving in bridging roles have been the focus of some social network studies [19]. Other research has examined the importance of formal mechanisms such as contracts [21,26]. Foster-Fishman developed a framework based on a qualitative review of the literature and identified four levels critical to build collaborative capacity: (1) member capacity, (2) relational capacity, (3) organizational capacity, and (4) programmatic capacity [27]. Strategies for building these capacities are provided and echo some of the factors listed above. For example, to build collaborative capacity, Foster-Fishman and others report that members’ belief in the need for collaborative action is necessary. However, processes or actions taken to achieve this state are not described in detail.

Most research on IOC has not focused on the contribution of successful IOC on health outcomes, but rather measures the process, such as improved collaboration or sustainability of networks [28]. This focus is problematic as network effectiveness at one level (e.g., the network) may be achieved at the expense of the other (e.g., the community or patient) [28,29]. Further, little research has focused on social network processes (e.g., how trust and shared vision is created and sustained) [30]. The history and evolution of coalitions vary significantly—some are top down and others are more grassroots. Such differences are likely to affect a coalition’s mission, authority, resources, and accountability. Some coalitions are visualized as temporary, as in a task force, while others are more permanent. Leadership and mission can change over time. Further, coalitions can often change the health systems and cultural contexts in which they work and are sometimes designed to do so as in the CFRs and OFRs. Detailed qualitative and longitudinal research is needed to help conceptualize important qualities and processes to optimize coalition work.

In this paper, we compare two multi-sectoral coalitions that are examples of IOCs to address opioid overdose in two counties. These coalitions were located in two very different contexts, had different lengths of existence, and had different levels of acceptance of harm reduction. We use these coalitions and programs as case studies to illustrate a framework to understand IOCs. In particular, we will examine strategies, activities, and processes that allowed coalitions to develop and function successfully, and barriers they encountered. We argue that coalitions are not static but continuously evolving and, in fact, must continuously evolve to remain relevant and effective.

## 2. Materials and Methods

Forty-four (44) semi-structured in-depth interviews were conducted with 46 key informants from 31 organizations across two counties in Wisconsin. As can be seen in Table 1, County A differed from County B in being urban and more racially/ethnically diverse than County B, which was relatively affluent and White. Overdose rates between the two counties also differed dramatically. Table 2 details the number of participants from various organizations by county. Purposive sampling recruitment methods were used to ensure organizations across the CoC were invited to participate. Members of the research team held pre-existing positions within overdose fatality review (OFR) teams and other overdose response initiatives, which assisted in the identification of potential research participants; some participants were known to interviewers prior to the interview as a result. We also recruited first responders, medical providers, and drug treatment providers who were not members of the OFR, although recruitment in some sectors, e.g., law enforcement, proved more difficult. All members of the research team were also strongly supportive of harm reduction efforts which may have introduced some bias in interviews with participants who were more ambivalent about harm reduction. Once a list of potential organizations in the two counties was developed, electronic communications were sent to organizational leaders inviting their participation. Included in the communication was information on the research project, the types of questions to be asked of participants, and information on incentives, as applicable. Incentives were not offered to full-time staff in government organizations as they were not allowed to accept incentives for participation in research during work hours. However, peer support specialists or other community health workers did receive a USD 35 incentive for their participation in a key informant interview. One participant who was contacted refused to participate, while four never responded to attempts to contact by email. Recruitment ended after all people on our expanded recruitment list had been invited to participate.

Research staff members who conducted key informant interviews included the female project coordinator (JO, MPA) who has many years of experience conducting substance use-related research and an advanced graduate student receiving her Ph.D. in Public and Community Health (SK). Interviewers asked participants to describe the overdose crisis in their communities, their organizations and roles within it, and different programs their organizations offer. In addition, we asked participants to talk about partnerships and coalitions that they formed with other organizations as part of their efforts to reduce opioid overdoses in their community including what had facilitated and impeded collaboration in the past and presently. Interviews were conducted virtually on a secure platform and lasted on average 40 min to 1 h. As mentioned above, the interviewers knew some of the participants prior to their interview from work with OFRs. Both were also strong supporters of harm reduction which may have introduced some bias although they matched participants in education and experience working with PWUO. However, the interviewers were trained to maintain a neutral and non-judgmental tone and expression in interviews. In addition, the research team frequently discussed the potential influence of this positionality on participant responses and our analysis and interpretation of findings.

All key informant interviews were recorded and transcribed verbatim. Transcripts were uploaded into MAXQDA software version 24, where coding and analysis occurred. Inductive and deductive codes were developed and used for analysis by the PI and interviewers [31]. Analysis occurred in three phases. The first phase involved content coding. Individual transcripts were read to identify the various types of overdose response programs and coalitions in the two counties. Next, we used a constant comparison approach and compared the findings between the two counties to identify factors that may explain similarities and differences, including length of time the coalition has existed and political context, among others. Lastly, we combined and integrated findings to draw generalizable implications and conclusions about interorganizational collaboration to improve overdose prevention. Deductive themes from the literature on building effective coalitions were brought in and applied or adapted as necessary, e.g., the importance of bridging people. All coding was performed collaboratively (JDG, JO, SK) and disagreements were resolved by consensus. Results were presented at the beginning of a 5-session group model building exercise with stakeholders who participated in the qualitative portion of the research. In these meetings, insights from interviews were used to identify key leverage points in the current opioid overdose response in County A.

## 3. Results

### 3.1. Differences Between Counties in Macro-Level Factors: The Extra-Organizational Landscape

The two counties studied were contiguous and included an urban center and a proximate county that included suburbs to the city and rural communities. Members of both coalitions reported engaging in various activities to “raise awareness” and reduce overdose stigma. These included community outreach activities, like naloxone training events. In many cases, the harm reduction potential of such events in terms of reaching people who would actually use naloxone was secondary to the opportunity to “educate” and change attitudes.

I think we have faced a lot of difficulty with like city image, because people have expressed concern about like, if we put out Narcan, they think it’s going to increase people using or misusing medications or drugs… So, it’s a little bit of a battle in some ways. I wouldn’t say it’s all been positive, it’s just a little bit more difficult. Like people don’t outright say anything, but they also are not super gung-ho about it either….(County A Participant 20, White male)

Participants reported that over time, these efforts to raise awareness have changed attitudes and the acceptability of harm reduction.

I think there’s some differing ideas as to what is best for our community or what our community would be receptive to. We’ve actually found that as well with our community Narcan training. We didn’t have as big of a turnout as we were hoping and some of the agencies involved with that training thought we should just cancel it… It was thrown around the idea of just doing a commercial Narcan training or posting a video on Facebook of how to use Narcan and—instead of kind of challenging the community and making them realize and accept that overdose is not—doesn’t discriminate against certain communities. It affects everyone. So, it’s just kind of working through that.(County A, Participant 6, White female)

Another strategy participants described to increase acceptance in sectors with missions unsupportive of harm reduction (e.g., police departments) was to invite and work with members of an adjacent sector, like fire departments.

When we first arrived, all of our more harm reduction overdose prevention efforts were met with a lot of resistance from the community, from first responders, even the fire department. We really quickly partnered with the [County A Fire Department] on the [overdose response] project… Which then, kind of once you have a partner in the first responder world that’s kind of subscribed to what you’re doing, then the other ones tend to [accept it], cause they talk, right? They’re all friends, they’re peers.(County B, Participant 18, White male)

Despite these commonalities, there were important differences in the outer context of the two counties. First, County A had been involved in overdose prevention for a longer time than County B. County A had an established OFR, with robust community engagement, which has evolved over time. County B, in many ways, was still forming its coalition and was even still searching for a name to capture its evolving mission.

So, the task force was in existence pre-COVID, right? So, it grew out of a concern within [County B’s] Health and Human Services around opiate and heroin issues and just trying to bring community members and the county government and all the partners together to try to effectively address the opiate and heroin issue. COVID caused a pause in it all. And so, it got restarted probably about a year, a year and a half ago now. And then kind of morphed our work and took a different strategy or a different approach than what the task force was doing before COVID… Our organization was the fiscal agent for the Drug Free Communities grant, so focused a lot on youth prevention and coalition building and those kinds of things… So, in rebooting the task force several of the people that were on the… design committee I had worked with… approached me personally about potentially becoming a leader for the task force in the new revised, updated [name of coalition].(County B, Participant 8, White female)

The newly rebooted task force was still defining its mission and membership. County B was in the process of beginning its own OFR when data were collected. If that happens, it will almost certainly affect the functioning of the current task force which may merge with the OFR or continue to exist independently, among many possible scenarios.

### 3.2. Forming Alliances: Diverse Group of Stakeholders to Address Multiple Needs of People with OUD

In both counties, the coalitions started with the realization that people with OUD had multiple, complex social and medical needs that would need to be addressed to decrease opioid overdose. Therefore, both coalitions sought a broad membership representing different sectors such as harm reduction, criminal justice, fire/emergency medical services, health, substance use, and the medical examiner’s office, among others. This was particularly the case for the County A OFR.

Participant: It is difficult to get all the people in the same room, because a lot of these people do have very busy schedules… We try to add people from like all different aspects of [name of city]. So, we have, like pharmacists. We have people from the mobile integrated health unit. We have a police lieutenant that helps like pick our cases. We have uh someone from the medical examiner’s office that talks about the autopsy report. In the toxicology findings we have a social worker from the medical examiner’s office. We review the cases in advance, and if they have any child involvement or any minors at all that could be in the school district we include the lead school social worker for [name of city] schools… anyone who has potentially had like been in touch or worked with the decedents in the cases.(County A, Participant 20, White male)

Regular meetings with members of different sectors allowed participants to understand other points of view which, in turn, led to more innovative solutions.

I think our taskforce, one of the things that we’ve done in the last year or year and a half, we’re spending quite a bit of time educating together, learning together, and building understanding about resources, as well as data… And we have all the sectors there. We have law enforcement, and we have healthcare, and we have therapists, and things like that. So, I think understanding where everybody starts and stops, and even questioning and pushing why can’t we do something a certain way.(County B, Participant 4, White female)

Having “everyone at the table”, however, does not necessarily mean that all voices are heard.

So, I think the Health and Human Services is very committed and they want to see success and so they have a very strong role in it, and they have access to resources and things that nobody else does, right, because they’re the county. I think they are really trying hard to figure out how do we engage their community partners. But I think there’s some power differential there like an organization like mine… I’m at the table, but I’m the only really, community partner at the design table… And so, having them at the table and having their ear, you can really potentially make things happen… The negative is that sometimes they are overpowering, and they forget that they’re not the sole solutions to the problems, right, and sometimes it’s hard to knock on the door and say, “Hello, we’d like to come in.”(County B, Participant 8, White female)

Having representation from multiple sectors was reported to be essential to “break down silos” that lead to fragmented services.

You also are seeing just both within the meetings and then outside of the meetings, the level of connection that’s happening between agencies and sometimes between individuals that may not have even had the opportunity to know the function of the other person or the other agency prior to some of this collaborative work and that can be really powerful. So, … we had the folks from the housing division come in and present in response to what we were seeing with one particular overdose incident, and their perspective, and then being able to connect them into this process. And knowing a little bit more about what they were seeing on their end really enhances our ability to all work together.(County A, Participant 21, White female)

Smaller social service agencies in particular saw value in attending meetings with state and county agencies to educate them on the services they provide. Since first responders, including police and fire, are often the first on the scene of a non-fatal overdose, educating them about harm reduction and other social services was a way to reach more people with OUD in crisis.

What’s I guess, what’s made it easier to basically having the conversations, you know, they don’t know what they don’t know. So, if a city doesn’t know you offer a service, or that you can provide a service, then they’ll never get on board. So obviously being part of OFRs… and all that is good, because you see who is in those meetings.(County A, Participant 1, Black male)

“Breaking down silos” was also seen as important to coordinate and remove the duplication of services.

Some barriers are that there’s a lot of people working in substance use prevention kind of independently. And so a barrier is getting people on the same track, not duplicating services, making sure we’re all moving in a direction together. We have definitely had that lesson in making sure that we’re not competing with each other for the same audience, trying to accomplish the same program.(County B, Participant 7, White female)

### 3.3. Creating a Shared Mission Helps Organizations to Collaborate

As might be inferred from some of the quotes above, coalitions representing such diverse stakeholder groups and institutions did not always see eye-to-eye or share organizational missions or values, e.g., harm reduction versus criminalization approaches to substance use. They all came together for the same purpose, to reduce opioid overdoses, but did not necessarily agree on how that would best be achieved. Part of the work of the coalitions, therefore, was to create a shared mission.

Interviewer: And how does your organization communicate with other organizations that are involved in overdose prevention?

Well, we hear from the participants or from… our community-based organization meetings provided by DHS, or from warm handoffs from somebody we may know through our networks. And we then discuss, these are important items to us. This is our mission for harm reduction, and we find a mutual understanding with these organizations, with their goals and what they anticipate with their programming, and we match it and if it’s parallel, we get together and we tackle those things together. And if they’re not parallel, what can we do to find some commonality to meet your mission and your goals regarding harm reduction and meet ours as well. Again, reminding you that silos is the death of prevention and the death of harm reduction.(County A, Participant 9, Hispanic male)

Some of the most basic negotiations and conversations that occurred were around attitudes and stigma toward people who use drugs and a mistrust in harm reduction. These were not one-time conversations.

Interviewer: Are there any barriers to those collaborations?

Yeah. Yes. So, police primarily. Which is understandable. Super frustrating, but they come at it from a law enforcement perspective, ‘cause it’s their job. And so, helping people to reduce the harm associated with their use, to them, can sometimes sound like allowing them to break the law, and being okay with them breaking the law… So, they’re much slower, I think, to come around and do… certainly resist with partnering with us on paper… And then, we have conversations with our partners, with first responder partners about the language that they use. It’s baked into the culture sometimes, that you use words like “Junky” or “Addict” or any of that kind of stuff that’s stigmatizing. And people don’t realize it’s stigmatizing until it’s been pointed out. And even then, there’s a little bit of resistance.(County B, Participant 18, White male)

Peer support specialists were particularly well positioned to offer feedback to partners to reduce stigmatizing language because of their lived experience. The participant above also seems to emphasize a gentle, non-confrontational approach. However, participants also talked about having champions, particularly those in leadership positions, to facilitate collaboration and to change behavior that might inhibit it, like stigmatizing language.

Attendance at shared community-facing events helped develop trust between organizations, helping also to build a sense of shared mission.

We are actively working and supporting one another’s efforts. For example, we go out every Monday. We had a meeting about a month ago in regards to what targeted areas do we need to go to, and we invited everyone to our end of the year celebration… And just because of the relationships that we have built this past year, everyone came out to support us there.(County A, Participant 27, Black female)

Just getting together regularly seems to help. I mean, we see each other now, at different events, there was an overdose prevention thing in the [name of park] that people had tables, and it was all the same groups. So it was, you know, good to see each other again… I mean building friendships really, among the people in those groups is really kind of what solidifies things.(County A, Participant 18, White male)

Creating a sense of shared mission is important not only to build trust and collaboration among organizations from very distinct sectors with different missions, values and cultures, but also sometimes helps to build trust in different organizations from the same sector. For example, healthcare organizations often feel as if they are competing for the same clients, making collaboration more difficult. As this participant expresses it, the current public health emergency means there is “no lack of participants”.

There’s a big need. There’s all these gaps in services, right? They exist everywhere. But especially between what would be seen as competing organizations, they’re all competing for the same clients, or whatever… There’s no lack of participants, right? And so, hopefully with the coalition, we’re trying to get everybody at the same table, and speaking the same language, and helping each other with the efforts… Having all these different partners kind of work together and pool resources is gonna be really, really helpful.(County B, Participant 18, White male)

### 3.4. Problem-Focused Discussions

OFRs are structured to review particular cases to identify potential points of intervention that may have prevented the overdose. This structure was particularly fruitful in terms of designing new programs or initiatives in County A which had active OFRs. Because of the broad coalition membership, it was then easy in the meetings to identify organizations that could provide needed services.

If we think there could’ve been a program that might have been able to intervene with this individual or prevent the overdose deaths, oftentimes an agency can say “Oh, we have that,” or “We have—we know someone who does that and can connect you” and it’s just that kind of setting brings—brings up a lot of resource sharing for agencies who are on those meetings…(County A, Participant 6, White female)

Identifying these potential areas of collaboration, the coalition then worked to support some of these with contracts, funding, or other resources.

As far as OFRs, it’s definitely helped us collaborate with other organizations. We’ve been working on getting some of those BAAs [business association agreements] in place. So, looking at the DOC. I think that’s how that one started was through a recommendation from an OFR which was great, and then looking at other mental health organizations… Those conversations have started because of OFRs, so again, you know, we break down those silos. We collaborate. We are finding out what works for folks. And then also it’s good to see what everyone else is doing. I think everyone has come from the standpoint of we have this north star of this, like mission of, preventing overdose, and I think that drives everyone’s passion and understanding.(County A, Participant 1, Black male)

### 3.5. Key People in Organizations Act as Bridges

Almost all participants mentioned key people who acted as bridges between different organizations, including people within participants’ organizations who were more outward facing and people from other organizations who served as “conveners”. These bridging individuals were also often the points of contact for outside organizations.

So, we have this collaborative [person’s name omitted] from [OTP], she um is that hub. So, every flyer, every new bit of information, goes to [person’s name omitted], and she’s the one that distributes. How she became that person, I don’t know. But she’s really good at it, and she does that.(County A, Participant 4, White female)

Because of the nature of their jobs helping people who use opioids to navigate various systems to access the services they may need, peer support specialists often become those people.

Yeah. I think the gaps are coordination, for sure. Just coordination of systems and agencies and organizations because it’s incredibly challenging for people to get through a system to get what they need because you just have to go all over the place. Each time you go someplace, you got to retell your story, you got to be reassessed, got to be this and that. We always talk about trauma-informed and person-centered care, we have the least probably trauma-informed system and person-centered system… The navigator [in our program] really helps that person connect all those pieces so they’re not in it by themselves. And if you walk with that person through this journey, they’re gonna be more successful than just pointing them in the right direction saying, “Go here.” You know?(County A, Participant 16, White male)

## 4. Discussion

This project qualitatively examined IOCs in community-wide interventions to reduce opioid overdose. Research continually points to the need for integrated medical care in improving health outcomes among populations suffering from complex medical problems [19,22]. OUD is a chronic condition that affects multiple aspects of the lives of people who suffer from it, including legal issues, damaged family relationships, unemployment, eviction, and mental and physical health [32]. Few agencies can address all the care that people with OUD may need to reduce the harms associated with opioid use. Therefore, it is important to study the conditions and processes that are associated with successful accomplishment of collaborators’ goals to maximize effectiveness of community-wide interventions. We had few specific hypotheses about IOC to guide data collection in this project. Nevertheless, participants’ experiences illuminated some strategies and actions that led to the success of IOCs in terms of identifying and addressing current gaps in the Opioid Continuum of Care. In both counties, organizations came together in coalitions (task force in County B, OFRs in County A) which helped them to collaborate with other organizations along the Opioid CoC. OFRs were found to be particularly helpful, however, in identifying gaps in the current CoC and developing programs in collaboration with other organizations to address them. These results would seem to support the value of funding coalitions to facilitate collaboration.

Figure 1 shows the beginnings of a framework to conceptualize the value of coalitions in fostering interorganizational collaborations and draws upon the organizational empowerment (OE) literature. Peterson and Zimmerman’s framework for OE includes intraorganizational, interorganizational, and extra-organizational components [33]. Coalitions help organizations connect with other organizations to prevent and respond to opioid overdoses effectively in a context that continues to stigmatize and criminalize people who use opioids [34]. Intraorganizational processes are those that facilitate organizations to participate effectively in coalitions. These included leadership support for coalition work and the importance of certain people acting as “bridges” to other organizations [35]. Participants also described how different organizations within coalitions often had different missions and practices that sometimes conflicted. In these cases, work within organizations to align missions to prevent overdose and resolve conflict was important to make harm reduction more acceptable to law enforcement officers.

Interorganizational processes are those which facilitate collaboration between organizations to facilitate a desired outcome [33]. Some of the most studied in OE include building alliances, getting the word out, and capturing others’ attention, all of which were applied by participants in this study [35]. Participants in both communities built alliances that included members of organizations from multiple sectors including law enforcement, medicine, harm reduction, and drug treatment, among others. People shared the programs their organizations offered that may have addressed decedents’ needs and prevented an overdose in OFRs, thereby “getting the word out” about their organizations’ missions and programs. “Capturing others’ attention” refers to the perceived importance of a problem among competing priorities. For the participants in this study, putting a human face on opioid overdose fatalities was a constant part of their message to increase the sense of urgency that they felt the problem deserved. IOC was fostered by activities to build a shared sense of mission and trust over time, in part a result of work together on coalitions.

At the extra-organizational level, success in these efforts to reduce overdose and successfully implement harm reduction can change laws and policies or create new programs. Participants counted many policy victories such as naloxone vending machines, media campaigns, and other efforts to reduce stigma around opioid use and increase acceptance of harm reduction [35]. As seen in this paper, many of the coalitions resulted in particular programs meant to prevent opioid overdoses including multi-disciplinary overdose response teams and expansion of naloxone distribution sites, among many others, as organizations learned about the work each did and found new ways of supporting each other.

Coalition members described spending considerable time and effort in educating the community about the benefits of harm reduction approaches in saving lives. The Community Readiness Model (CRM) may be a useful tool to amplify and systematize some of the work communities are already engaged in to persuade the community to support harm reduction efforts [36]. The CRM assesses communities along a set of nine stages of a community’s “readiness” to address a public health issue ranging from “no awareness” of a problem to “professionalization”. However, coalition members in this project did not feel the need to engage in education to raise awareness of opioid overdose as a problem. This is noted in the model as a “sense of urgency” which was already rather high in the community. Rather, their focus was on convincing people that harm reduction was an effective solution to many of the harms caused by substance use, much of which is a product of the criminalization of drug use in the country. The CRM stages could be adapted to reflect this somewhat different problem, but more research is needed on ways to effectively change stigmatizing attitudes toward harm reduction and substance use disorders [37]. Finally, as noted in the introduction and as pointed out by participants, funding sources could either encourage or discourage IOC. In some cases, participants felt that they were in competition with each other for funding for similar programs. More often, however, funding was sought in partnership and helped to facilitate coalition building.

Participants in both counties expressed that harm reduction is still not well accepted everywhere, and that more politically conservative districts were particularly opposed. In addition to setting the overall climate, some coalition members may be in elected positions, such as sheriffs, complicating public health decision making. Appointed officials may also receive pressure from elected officials to make decisions based on popular sentiment rather than public health, similar to what was seen in some areas of the US during the COVID-19 pandemic [38]. While no real organized campaign against harm reduction occurred, there was less support for harm reduction in County B compared to County A—as one participant remarked, officials did not oppose any actions, but they were not “gung-ho” either. Another participant reported that having elected people on coalitions was good because they had the power to make policy changes or develop programs—of note, this participant also complained that power differences can crowd out community voices. Implicit bias training, positionality, and structural competency training may help to address some of these power imbalances that can affect equitable partnerships. However, it is important to understand that not only did coalition members work within a particular political context, but they were also actively involved in working to change that context. Participants described a great deal of their work as “educating”, “raising awareness”, or reducing the stigma associated with opioid use and harm reduction. One can imagine that the mission of these coalitions will change as harm reduction materials become more widely accepted, distributed, and available.

Participants talked about how the structure of coalitions allowed people from organizations with very different missions and purposes to reach common purposes and create and expand overdose response programs. The coalitions in both counties were purposeful in recruiting members to represent these diverse sectors and were continuously evolving over time. Participants also described actively working to create a sense of a shared mission with other organizations and the people working in them. Processes to achieve these included finding a common language to talk about the problem and finding goals in common even when missions do not completely align. Coalition members also talked about creating a sense of trust over time by participating in community events together or through positive interactions when referring or receiving clients from these other organizations. It is important to note that IOCs will have varied experiences with developing a sense of mission and mutual trust, likely due to their distinct histories and rationales for their existence. Coalitions help establish a common purpose among organizations, but trust is also affected by these day-to-day interactions within IOCs.

As seen in much research on IOC, participants identified bridge people as important in helping organizations to provide integrated services to communicate. These bridge people can be described as having high “social capital”; in other words, they are people who can mobilize resources and social relationships for social ends [39]. Peer support specialists often played that role as they helped people with OUD navigate the system to address their multiple needs while still having trust of people who use drugs. Despite their critical role, there have been many challenges in integrating PSS into different organizations. Organizational policies, such as abstinence requirements [40], limit who may qualify as eligible to hold a peer support specialist role. Limited public and private insurance reimbursement and funding of peer support prevent the sustainability of the roles [41,42] and restrict the number of peers that organizations can hire. Despite integration challenges, peer support specialists are preferred by clients [43], and increase the number of referrals to services for people with OUD [42,44], the number of appointments kept [45], and the retention in treatment [40,45,46,47].

Participants from both counties talked about the importance of having representation from diverse sectors given the complexity of the problem. However, OFRs seemed to offer the additional advantage of being able to identify gaps in services and policies to develop programs that would address these. All who had participated in OFRs could describe at least one example of a program or collaboration that had developed from the reviews. Research on the implementation of OFRs and their effectiveness in reducing opioid overdoses within a community is still relatively limited. Most existing research has focused on implementation practices [48,49], but our research suggests that they can be quite effective in increasing programming and funding for overdose prevention. If supported by the research, OFRs could be promoted by funders by requiring their establishment as part of community overdose prevention efforts.

Limitations of this study include the small sample size in two very specific local contexts, and the non-generalizability of the qualitative research. In addition, this project included only one in-depth interview per participant, limiting the ability to study changes over time. Future longitudinal research is needed to focus on evolution and change in coalitions for health over time. Such research could add insight by combining qualitative and quantitative methods, including social network analysis.

## 5. Conclusions

The opioid overdose crisis requires IOC. Case studies can provide information that can then be used to develop best practices that are flexible enough to be used in diverse contexts and with other complex, chronic conditions.

## Figures and Tables

**Figure 1 ijerph-22-01765-f001:**
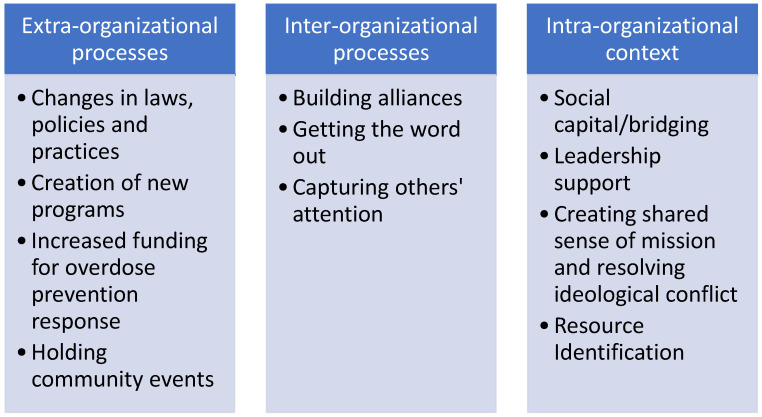
Organizational empowerment.

**Table 1 ijerph-22-01765-t001:** Comparison of study sites.

	Population	Race/Ethnicity	Median Income	Urbanicity	Overdose Rate 2023
County A	961,205	White 49% Black 26% Hispanic 16%	USD 62,000	Urban	58.2/100,000
County B	406,978	White 87% Black 1.7% Hispanic 5.4%	USD 88,985	Suburban/rural	14.8/100,000

**Table 2 ijerph-22-01765-t002:** Number of participants in key informant interviews in each county by organization type.

	County A	County B
Academia **	2	0
Criminal Justice	2	2
Department of Health	4	2
EMS	4	1
Government	3	0
Harm Reduction	3	7
Other Social Services	1	2
Peer Support Specialists	3	2
SUD Treatment	6	4
Total # of participants	27	19

Note: Some key informant interviews included two or more people, which accounts for the variation in the number of interviews versus the number of participants. ** The two key informants from Academia are the facilitators of the urban OFR.

## Data Availability

The data presented in this study are available on request from the corresponding author due to confidentiality concerns.

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
