# Peer review of "Strategies to Facilitate Interorganizational Collaboration in County-Level Opioid Overdose Prevention and Response: A Qualitative Analysis"

_ijerph, 2025, doi:10.3390/ijerph22121765_

Round 1

Reviewer 1 Report

Comments and Suggestions for Authors

This is a really interesting paper. I think the paper would benefit from some literature pertaining to stigma - I would recommend Michelle Addison's research on this. In your methods section, there is no literature to support and clarify your approach. I would suggest using citations for purposeful sampling and for which analysis method you used. I noted that you talked about deductive and inductive themes, and have assumed that you have used Braun and Clarke's reflective thematic analysis, but this is not clear. It would be useful to clarify what the deductive theme's were within this section. I appreciated that there was acknowledgement of researcher bias and participant bias. 

For the findings section I would suggest that you start with outlining the themes you are going to discuss and clarifying whether there were further themes that analysis identified that you have opted not to discuss in the paper.  It is quite difficult to read the findings section because of the formatting. For example, participant quotes are not indented nor are italics/quote marks used. So I suggest some formatting improvements. I noted that a pseudo name had been used for Faith - and wondered if it would be better to simply put in [persons name omitted]? With the name Faith having religious connotations, you do not know if that would offend the actual person named?  

Overall, I appreciated the paper and found it interesting and I think it offers a unique contribution to the field. I sit on various partnership boards in the UK where we are working together to address drug related deaths as one of our targets. There can be challenges with organisation's having differing ethos and approaches - particularly criminal justice versus public health approaches. 

There were a few referencing improvements that I spotted and double spaces between words - so a general edit and referencing check is advised. 

Overall, a really well written and informative paper.  

Reviewer 2 Report

Comments and Suggestions for Authors

This paper – describing a qualitative study aiming to understand the facilitation of better interorganizational collaboration in the context of addressing local opioid overdose prevention and response – is very well-written.  Largely, the writing is very clear and thorough.  I have a few suggestions, that in my view, will help strengthen the paper even further, if addressed.

  1. Readability and flow will be improved if the interview transcriptions presented are set apart more explicitly.
  2. How many participants endorsed each theme? It would be helpful to include a small table that indicates the ns for each theme.  Alternatively, it might be easier and more efficient to simply include (n = #) at the end of each theme heading.
  3. Were there any minor themes?  If so, a brief description and discussion would be useful.

Reviewer 3 Report

Comments and Suggestions for Authors

Round 2

Reviewer 3 Report

Comments and Suggestions for Authors

The changes have been carefully and completely handled. The manuscript is now coherent, transparent in its methodology, and in line with the Organizational Empowerment concept. In my opinion, the revised version seems suitable for publication.